# Molecular Insights into the Multifunctional Role of Natural Compounds: Autophagy Modulation and Cancer Prevention

**DOI:** 10.3390/biomedicines8110517

**Published:** 2020-11-19

**Authors:** Md. Ataur Rahman, MD. Hasanur Rahman, Md. Shahadat Hossain, Partha Biswas, Rokibul Islam, Md Jamal Uddin, Md. Habibur Rahman, Hyewhon Rhim

**Affiliations:** 1Center for Neuroscience, Korea Institute of Science and Technology (KIST), 5 Hwarang-ro 14-gil, Seoul 02792, Korea; 2Global Biotechnology & Biomedical Research Network (GBBRN), Department of Biotechnology and Genetic Engineering, Faculty of Biological Sciences, Islamic University, Kushtia 7003, Bangladesh; mrislam@btge.iu.ac.bd; 3ABEx Bio-Research Center, East Azampur, Dhaka 1230, Bangladesh; hasanurrahman.bge@gmail.com (M.H.R.); shahadat4099@gmail.com (M.S.H.); parthabiswas2025@gmail.com (P.B.); hasan800920@gmail.com (M.J.U.); 4Department of Biotechnology and Genetic Engineering, Bangabandhu Sheikh Mujibur Rahman Science and Technology University, Gopalganj 8100, Bangladesh; 5Department of Biotechnology and Genetic Engineering, Noakhali Science and Technology University, Noakhali 3814, Bangladesh; 6Department of Genetic Engineering and Biotechnology, Jashore University of Science and Technology, Jashore 7408, Bangladesh; 7Department of Biotechnology and Genetic Engineering, Faculty of Biological Sciences, Islamic University, Kushtia 7003, Bangladesh; 8Graduate School of Pharmaceutical Sciences, College of Pharmacy, Ewha Womans University, Seoul 03760, Korea; 9Department of Global Medical Science, Wonju College of Medicine, Yonsei University, Seoul 03722, Korea; pharmacisthabib@gmail.com; 10Division of Bio-Medical Science and Technology, KIST School, Korea University of Science and Technology (UST), Seoul 02792, Korea

**Keywords:** autophagy, cancer, phytochemical, natural compound, treatment

## Abstract

Autophagy is a vacuolar, lysosomal degradation pathway for injured and damaged protein molecules and organelles in eukaryotic cells, which is controlled by nutrients and stress responses. Dysregulation of cellular autophagy may lead to various diseases such as neurodegenerative disease, obesity, cardiovascular disease, diabetes, and malignancies. Recently, natural compounds have come to attention for being able to modulate the autophagy pathway in cancer prevention, although the prospective role of autophagy in cancer treatment is very complex and not yet clearly elucidated. Numerous synthetic chemicals have been identified that modulate autophagy and are favorable candidates for cancer treatment, but they have adverse side effects. Therefore, different phytochemicals, which include natural compounds and their derivatives, have attracted significant attention for use as autophagy modulators in cancer treatment with minimal side effects. In the current review, we discuss the promising role of natural compounds in modulating the autophagy pathway to control and prevent cancer, and provide possible therapeutic options.

## 1. Introduction

Autophagy is triggered by the entrapment of abnormal intracellular proteins, invading microorganisms, and damaged organelles with the formation of double-layer autophagosomes, in the presence of nutrient stress, injury, and fasting [1,2]. Along with cellular energy stability, autophagy contributes to the control of cellular quality, and destruction of abnormal proteins and damaged organelles [3]. As a result, the origin and development of many diseases, such as neurodegenerative disorders, cancer, and autoimmune diseases, can be explained by defective autophagy [4]. Multiple data states a connection between the variety of diet and feedback to cancer medication [5]. It has been established that sufficient consumption of dietary and medicinal plant-derived biochemical compounds lowers the frequency of cancer mortality. This occurs due to the activation or balancing of various cellular oncogenes. In different human disorders, including cancer, there is documented evidence of reduction of autophagic regulation. Several lines of evidence have shown that cancer plays a critical role in inhibiting or augmenting autophagic pathways [6]. In addition, this dual role of autophagy has a significant therapeutic benefit against cancer [7]. The present report has established that natural product-derived bioactive molecules, along with lysosomal inhibitors, including chloroquine (CQ) and hydroxychloroquine (HCQ) are important regulators of autophagy signaling [8]. In addition, these proactive molecules can regulate the process of autophagy both in vitro and in vivo by involving the enzymes, transcription factors, and various intracellular communication pathways [9]. The autophagy stimulation or prohibition process is vastly complicated and regulated, so it needs to be extensively explored. However, emerging evidence has implicated that extreme or damaged autophagy might lead to a unique type of cell death known as autophagic cell death [10]. Conversely, the proposal of using natural compounds as anti-cancer and autophagy-modulating agents requires a better understanding of their cellular and molecular mechanisms. The mechanism of action needs to be further addressed.

In this review, we compiled the cellular and molecular mechanisms involved in the dual role of autophagy, as a tumor-suppressing as well as a tumor-augmenting phenomenon in cancer. Current improvements in the use of natural bioactive compounds for cancer treatment by targeting autophagic action have been consistently analyzed [11]. Given the vital role of autophagy in cancer prevention and treatment, we closely reviewed different plant-derived biomolecules that may play a role in modulating the autophagy-linked signaling pathways as well as contribute towards competent treatment strategies against specific molecular aspects in cancer. In particular, administration of naturally derived compounds have been considered to regulate autophagy modulation, which might help improve our understanding of the mechanisms of natural compounds in the management and treatment of autophagy-linked diseases and cancers.

## 2. Mechanisms of Autophagy Signaling Pathway

Autophagy is a sensitive and tightly mannered process that has multiple steps [12]. Proteins crucial for autophagy were first reported in yeast and are termed as autophagy-related (Atg) proteins [1,13]. Multiple sequential molecular events take place to initiate the autophagy signaling cascade. First, an Atg1/unc51-like kinase (ULK) complex kinase regulates the induction, and the BECN1/class III PI3K complex initiates the nucleation of vesicles [14]. In addition, ubiquitin-like conjugation systems (Atg12 and Atg8/LC3B) control vesicle elongation as well as retrieval of lipids mediated by transmembrane Atg9 and associated proteins, which involve other Atg proteins [15]. Lysosome-associated membrane protein 2 (LAMP2) and RAS-related protein-7 (RAB7A) participate in the fusion between lysosomes and autophagosome formation [16]. Finally, vesicle breakdown and degradation by lysosomal hydrolases produce recyclable products [17] (Figure 1). Microtubule-associated protein 1 light chain 3-beta (MAP1LC3B) and Atg5-Atg12 complex pathways, which are ubiquitin-like pathways, are regulatory for vesicle elongation and are encoded by the yeast Atg8 pathway, which has a human homologue [18]. Following the autophagy response, human Atg4 cysteine protease cleaves LC3B protein and thereby produces the LC3B-I isoform [19]. Activated LC3B-I isoform causes the activation of Atg7 protein through LC3B-I and is transported to the Atg3 before formation of the LC3B-II isoform by the phosphatidylethanolamine (PE) conjugation with the carboxyl glycine of LC3B-I protein [20]. As a consequence, elevated synthesis and processing of LC3B shows promise and has been mentioned as a key marker of autophagy [20]. In the degradation of matured autophagosomes through the enzymatic action of lysosomal enzymes, small molecules are produced that are used for de novo protein synthesis, helping in survival and maintenance of homeostasis [21].

## 3. Molecular Mechanism of Autophagy Signaling in Cancer Pathogenesis

Under healthy conditions, cells have an innate ability for autophagy mechanisms to defend against malignant transformation [22]. Generally, autophagy is initiated by the induction of tumor-suppressing proteins, and oncoproteins are reported to inhibit this action [23]. Therefore, autophagy can act in both pro-survival and pro-death stages of tumor initiation and development [23]. The disturbed autophagy pathway can contribute to tumor development, as it leads to accumulation of damaged organelles and protein aggregates, which ultimately produce reactive oxygen species (ROS) and lead to genome instability [24]. Several researchers have reported that chemotherapeutics and avenues for cancer treatment can be accelerated by altering autophagic signaling that result in tumor cell death by inhibition of pro-survival and tissue specific apoptosis inducing factors [25]. Additionally, autophagy has been described as having a dual function in the promotion as well as inhibition of metastasis [26]. In the early stages, autophagy prevents metastasis, whereas in later stages it favors metastatic activity [26]. However, poorly vascularized tumor cells may survive at low nutrient and low oxygen levels through TGF-β, and other signals that trigger autophagy [27]. Furthermore, autophagy has directly regulated the metastatic cascade mainly through migration, invasion, and epithelial to mesenchymal transition (EMT) of cancer cells [26,28]. The importance of autophagy at diverse stages of tumorigenesis and metastasis is illustrated in Figure 2. It has been found that many cells within the tumors such as immune cells, fibroblasts, and endothelial cells are surrounded by the tumor microenvironment and have been acknowledged as an attractive target to reduce resistance to anticancer treatment. Natural compounds from vegetables, spices, marine organisms, herbs, and fruits have been reported to prevent or reverse multistage carcinogenesis as well as to prevent cancerous cell proliferation [29]. Additionally, microRNAs (miRNAs), which control the expression of genes, play an important role in diverse biological processes. It has been found that miRNA expression dysregulation is highly related to cancer development [30]. Recently, several studies have shown that natural compounds, such as paclitaxel (PTX), genistein, curcumin, epigallocatechin-3-gallate (EGCG), and resveratrol exhibit pro-apoptotic or anti-proliferative properties that are controlled by miRNAs, and lead to inhibition of cancer cell growth and proliferation, apoptosis induction, and improvement of conventional cancer treatment and therapeutic efficiency [30]. However, these naturally occurring compounds could be modulated by different signal transduction pathways through their connection via cancer cells, tumor microenvironment, and miRNA. These are described later in this study.

### 3.1. Autophagy Signaling Roles in Cancer Inhibition

Several studies have reported that autophagy inhibition via manipulated pharmacological or genetic tools can be a possible chemotherapeutic option. Recently, scientists have focused on the combined treatment of chemically synthesized autophagy inhibitors and traditional chemotherapeutics [31]. Cancer cell sensitization with 3-methyladenine (3-MA), wortmannin, CQ, LY294002, bafilomycin A1 (Baf A1), and HCQ-like inhibitors of autophagy to chemotherapeutic drugs have been reported to accelerate cell death pathway action [32,33]. In the abovementioned inhibitors, some of them are preliminary stage inhibitors of autophagy, like wortmannin, LY294002, and 3-MA, which block the autophagosomes formation, while others act on the basis of lysosomal action at late stages [34]. Table 1 shows several valuable effects of natural compounds combined with some autophagy inhibitors in different cellular cancer models. In breast cancer cells, the pro-survival nature of cancer cells can be downregulated by LC3 via shRNA blocking and sensitization of the carcinoma cells to apoptotic factors such as trastuzumab [35]. In leukemia, pro-survival function with pharmacological inhibitors or RNAi can also block accelerated induced cell death by imatinib mesylate [36]. Autophagy can sensitize and make cells susceptible to radiotherapy, such as HBL-100 cells, for radiotherapy treatment, and 3-MA leads to the inhibition of autophagy by imposing the pro-death effects of therapy [37]. In BIF-1 knockout mice, there is impairment of autophagy in addition to an increase in the rate of tumor formation, while mutations in UVRAG decrease autophagy along with increased proliferation of colorectal cancer cells [38]. These reports propose that Beclin-1 acts as an insufficient tumor-suppressor gene and fully incorporates the theory of tumor-suppressive autophagy induction at early stages [39]. In the livers of mice, Atg gene deletion can lead to the formation of multiple benign tumors. In systematic mosaic Atg5 and Atg7 deletion mice, these specific deletions magnify the chance of tumor induction and lead to tumor development [40]. At early stages of cancer formation, Atg4 suppresses tumor developing activity and Atg4c deficiency is proven to be more easily identifiable in chemical carcinogen-induced fibrosarcoma. These reports help to understand the suppressive roles of autophagy in cancer with genetic deletion of specific regions [41].

### 3.2. Autophagy Signaling Roles in Cancer Promotion

Autophagy induction is a possible anti-cancer mechanism in which cancer cells have a faulty or altered apoptotic cell death pathway [52]. Consequently, autophagy induction along with autophagy inhibition can induce apoptosis and exert anti-cancer effects [53]. Several inhibitors, such as Bcl-2, EGFR, and mTOR, are currently used in cancer treatment [54]. Inhibitory molecules are known to be effective against a variety of cancers, and some induce autophagy in breast cancers, gliomas, and lung cancers through rapamycin, an inhibitor of mTORC2 [54]. Another known inhibitor of the mTOR, deforolimus (AP23573, MK-8669), is proposed to treat patients with relapsed or refractory hematologic malignancies [55]. Another antibody against epidermal growth factor, cetuximab, can regulate autophagic cell death in cancer cells by disrupting the interaction between Bcl-2 and Beclin-1 [56]. Several reports explain that the expression of the viral oncogenes Kirsten and Harvey RAS virus is capable of autophagy induction in certain cells [22]. In pancreatic and colorectal cancer cells, elevated levels of RAS mutations are reported to be correlated with higher autophagy rates, where increased levels maintain cell proliferation in RAS-activated tumors and inhibition of autophagy results in reduced cell proliferation, ultimately leading to tumor regression [57]. Lung cancers are also dependent on autophagy, which is similar to the mechanism of RAS-driven cancers regulated by valine-to-glutamic acid substitution at BRAF position 600 (BRAFV600E) [58]. In vivo studies have reported that deletion of subunit FIP200 related to the ULK1 complex in autophagy initiation blocks the growth of breast cancer and extends the life cycle of mice [59]. In addition, activated AMPK generated by ATP can also trigger an autophagic event [60].

## 4. Natural Compound Triggers in Autophagy Modulation

A single therapeutic strategy and a combination modality can both be favorable for cancer treatment. The various strategies include radiation therapy, immune or genetic therapy, and chemotherapy, all of which create new diversions and avenues in the field of cancer treatment with adequate levels of safety and accuracy [15,61,62]. Natural compounds are effective in the positive regulation of anticancer activity, as they inhibit cellular proliferation, adjust the oxidative stress response, and manipulate autophagic signaling and response [15,63]. This entire response of natural compounds is dependent on the cell type. The role of several natural compounds that modulate different pathways of autophagy signaling is illustrated in Figure 3.

### 4.1. Inhibition of Autophagy by Natural Compounds for Cancer Therapy

Apigenin: Apigenin, a flavonoid found in vegetables and fruits, can initiate tumor cell death through autophagy and apoptosis [65]. In combination with retinamide, *N*-(4-hydroxyphenyl), apigenin suppresses autophagy and increases apoptosis in human neuroblastoma (NB) cells [66].

Genistein: Genistein, a nutritive natural compound, has been documented to repress autophagy and induce apoptosis of human colon cancer HT-29 cells alone with downregulation of the PI3K/Akt signaling pathway [67]. Another mechanism of action of genistein is that it also exerts an antiproliferative effect on ovarian cancer cells through autophagy as well as apoptosis [67]. Genistein augments miR-451 expression and improves isoproterenol-mediated cardiac hypertrophy [68]. Furthermore, genistein increases miR-1469 expression and prevents Mcl-1 expression in laryngeal cancer [69].

Indole-3-carbinol: Indole-3-carbinol, a natural compound collected from cruciferous vegetables, usually inhibits the PI3K/Akt pathway along with blockage of autophagy signaling in human colon cancer HT-29 cells [67].

Plumbagin: Plumbagin is obtained from *Plumbago zeylandica* L. root and has a prompt antiproliferative activity through autophagic cell death, mediated via inhibition of AKT/mTOR pathway in human cervical cancer cells [70].

### 4.2. Induction of Autophagy by Natural Compounds as Potential Cancer Therapy

Antroquinonol: Antroquinonol, produced from *Antrodia Camphorata*, shows anti-tumor activity in various cancer cells [71]. Antroquinonol has been demonstrated to exhibit inhibitory effects on non-small cell lung cancer (NSCLC), as well as human pancreatic cancer, PANC-1 and ASPC-1, via the PI3K/Akt/mTOR pathway [72]. The anti-proliferative effect of antroquinonol is related to apoptosis, autophagic cell death, and rapid senescence of the pancreatic cells [72]. Several studies have demonstrated that antroquinonol can be introduced for the treatment of neoplasms [73]. A proposed clinical trial in metastatic pancreatic carcinoma is aimed at evaluating the anticancer activity of antroquinonol in combination with nab-PTX and gemcitabine [74].

18α-Glycyrrhetinic acid: 18α-Glycyrrhetinic acid (18-GA), a known gap-junction inhibitor, has been shown to demonstrate anticancer properties in human NB cells [42]. 18-GA has been shown to induce autophagy as well as apoptosis. Autophagy increases Atg5, Atg7, and LC3II along with degradation of p62 (Figure 4). Furthermore, 18-GA, along with the autophagy inhibitor CQ, induces significant cell death in NB cells. Moreover, the Bcl-2/Beclin-1 interaction in addition to the cleavage of Beclin-1 has been revealed by 18-GA in autophagy and apoptosis induced cell death. Caspase-3 siRNA and pan-caspase inhibitor treatment have been demonstrated to prevent 18-GA-induced cellular cytotoxicity, indicating that caspase-mediated apoptosis induction was observed in human NB cells [42].

Celastrol: Celastrol, a popular natural medicinal compound triperine secreted from *Tripterygium wilfordii* Hook, is a polyubiquitinated aggregate that degrades the autophagy substrate p62 in the human glioblastoma (GBM) cancer cells [75]. Celastrol-mediated cytotoxicity was not sensitized by adjustment with autophagy inhibitors. Paraptosis-like cytoplasmic vacuolization can be induced by celastrol and has been associated with autophagy and the initiation of apoptosis in PC-3, A549, and HeLa cancer cells [76]. Celastrol can be used to treat fever, joint pain, and edema without causing any side effects [77]. Recently, many research studies have identified celastrol as a neuroprotective agent via a collaborative drug screen that can be used for the treatment of many neurodegenerative diseases.

Monanchocidin A: An active novel alkaloid, monanchocidin A, has been recently isolated from *Monanchora pulchra*, a marine sponge [78]. Monanchocidin can initiate autophagy and lysosomal membrane permeabilization in tumor cells [78]. Many studies have demonstrated that monanchocidin A can inhibit human urogenital cancers, including germ cell tumors, through autophagy signaling [79].

Paclitaxel: PTX, a natural medicinal compound isolated from *Taxus brevifolia* Nutt, shows a vast range of effect against various types of cancer such as breast, endometrial, bladder, and cervical cancer, by influencing the autophagy pathway [80]. The US Food and Drug Administration (FDA) has approved PTX for the treatment of ovarian cancer and early stage breast cancer. The anticancer activity of PTX is increased when PTX is combined with cisplatin or carboplatin, and the combined compound can be used for the early treatment of advanced ovarian cancer [81]. It has been shown that PTX can inhibit the initial stage of autophagy, along with apoptosis, under normoxic and hypoxic conditions in human breast cancer cells [82]. PTX-initiated apoptosis was combined with the initiation of autophagy in A549 lung cancer cells [47]. Treatment with PTX also initiated acidic vesicular organelle formation, Beclin-1, Atg5, and LC3 expression, and the prevention of autophagy proceeds via PTX-mediated cell death [83,84]. Autophagy can promote PTX-mediated cell death. In human breast cancer cells, PTX prevents autophagy via an individual mechanism based on the cell-cycle phase [84].

Gintonin: Gintonin (GT), a novel ginseng-derived exogenous ligand of lysophosphatidic acid (LPA) receptors, has shown to induce autophagy in cortical astrocytes [45]. GT intensely augmented the autophagy marker LC3 via the G protein-coupled LPA receptor-mediated pathway. However, GT-mediated autophagy was considerably decreased via autophagy inhibition 3-MA as well as Beclin-1, Atg5, and Atg7 gene knockdown [15,45]. Notably, pretreatment with a lysosomotropic agent, Baf A1 and E-64d/peps A, GT significantly improved LC3-II levels in addition to the formation of LC3 puncta (Figure 5). Furthermore, GT treatment improved autophagic flux, which influenced lysosome-associated membrane protein 1 (LAMP1) in addition to degradation of the autophagy substrate ubiquitinated p62/SQSTM1 protein in mouse cortical astrocytes [45].

β-elemene: β-elemene, derived from *Rhizoma Curcumae,* is a natural terpenoid bioactive compound that shows strong activity against various types of cancers [40]. In the human body, β-elemene works against the PI3K/Akt/mTOR/p70S6K pathway that induces autophagy and apoptosis of human NSCLCA549 cells [85]. As β-elemene increases the antitumor effect, autophagy is prevented by the interception with chlorochine [85]. In human renal-cell carcinoma 786-0 cells, β-elemene is used for the inhibition of the MAPK/ERK and P13K/Akt/mTOR signaling pathways to initiate defensive autophagy and apoptosis [86].

Caffeine: In HeLa cells, the Akt/mTOR/p70S6K pathway initiates autophagy and apoptotic cell death under the influence of caffeine that is routinely consumed in beverages [87].

Curcumin: Curcumin is a non-toxic toxic polyphenol with yellow pigment isolated from the rhizome of *Curcuma longa* L. [88]. Curcumin-associated autophagy is considered to be a signal for cellular death in various types of cancer cells in the human body. Nevertheless, in the human body, curcumin can induce cellular differentiation and cellular survival by regulating the activities of AKT-AMPK-mediated autophagy induction [89]. Thus, it prevents the growth of malignant glioma cells in vitro and in vivo. Curcumin is used to induce non-apoptotic autophagy cell death in U87-MG and U373-MG malignant glioma cells [90]. This activity is related to the G2/M phase cell cycle attack, the prevention of the ribosomal S6 protein kinase pathway with the ERK1/ERK2 activation pathway [90]. Curcumin is responsible for Atg apoptosis in mesothelioma and k562 long-standing myelogenous leukemia cells by AKT/mTOR and NF-κB signaling pathways [91]. Curcumin can be used as an effective therapeutic compound in cancer management and is well endured, as confirmed by preclinical data in clinical experiments [88]. Curcumin has been found to suppress TGF-β expression, such as p-SMAD2, TGF-β3, MMP-13, and NF-κB tumor-promoting factors as well as metastatic active adhesion molecules, such as intercellular adhesion molecule 1 (ICAM-1) and β1-integrin in stromal fibroblasts and colorectal cancer cells [92]. In addition, MiR-1246 has been shown to be involved in curcumin-induced radiosensitizing actions on bladder cancer cells by targeting p53 translation [93].

Tetrahydrocurcumin: Tetrahydrocurcumin (THC) is one of the strongest bioactive natural metabolite derived from curcumin. THC effectively shows strong antioxidant, anticancer, and cardioprotective properties [94]. The action of THC on autophagy was assessed by the decrease in the PI3K/Akt-mTOR and MAPK signaling pathways and initiation of caspase-7 mediated apoptosis [95].

Oxyresveratrol: Oxyresveratrol (Oxy R), 4-[(E)-2-(3,5-dihydroxyphenyl) ethenyl] benzene-1,3-diol/ 2,3,4,5-tetrahydroxy-trans-stilbene found in *Morus alba*, has been shown to activate autophagy and apoptotic cell death in NB cells [51] (Figure 6). Oxy R-mediated autophagic cells were independent of apoptotic cell death. The PI3K/AKT/mTOR and p38 MAPK pathways control Oxy R-induced cell death in SH-SY5Y human NB cells [51].

Epigallocatechin-3-gallate: The activity of cisplatin and oxaliplatin-initiated autophagy was promoted by EGCG in HT-29 and DLD-1 colorectal cancer cells, as identified by the multiplication of LC3B-II protein and autophagosome organization [96]. EGCG initiated autophagy as well as apoptosis in SSC-4 squamous cell carcinoma in combination with the upregulation of FAS, BAK, BAD, IGF-IR, WNTll, and ZEBI proteins along with downregulation of MYC, TP53, and CASP8 proteins [97]. The formation of autophagosomes is increased by EGCG. In hepatic cells, the processes of lysosomal acidification and autophagic flux are also increased by EGCG in vitro, as well as in vivo [98]. Remarkably, luteolin and EGCG combination repressed TGF-β-mediated demonstration of myofibroblast phenotypes by reducing RhoA as well as ERK activation [99]. EGCG suggestively repressed rat hepatic stellate cells proliferation through hindering expression of PDGF-β receptor and tyrosine phosphorylation [100]. Furthermore, it has been found that EGCG attenuates uric acid-mediated NRK-49 F cell injury by up-regulating miR-9, and successively by JAK-STAT as well as NF-κB signaling pathway activation [101]. EGCG reduces carcinoma cell growth probably by regulating miRNA expression, and may be a potential beneficial target for the prevention of cancers.

Fisetin: Fisetin (3,7,3′,4′-tetrahydroxyflavone), a flavonol and a member of flavonoid polyphenols [102], is usually isolated from many vegetables and fruits [102]. Fisetin can function as an inhibitor of the expression of PI3K/Akt/mTOR pathways and regulate autophagy in prostate cancer and human NSCLC cells [103,104]. Fisetin has been found to inhibit the mTOR pathway and induce autophagy in human prostate cancer cells [105]. In addition, the effects of fisetin on autophagy are cell-type specific because fisetin inhibits autophagy in MCF-7 breast cancer cells and induces caspase-7-mediated apoptosis [106].

γ-Tocotrienol: Tocotrienols and the isoforms of vitamin E are differentiated on the basis of their antioxidant, anti-inflammatory, and anticancer properties. γ-tocotrienol induces endoplasmic inflammation and autophagy-associated cell death [107]. Pretreatment with the autophagy inhibitors 3-MA or Baf1 prevented the γ-tocotrienol-initiated cytotoxicity [108]. Many other studies have demonstrated that in mouse mammary cancer cells, the attachment of γ-tocotrienol with oridonin effectively initiated both autophagic and apoptotic effects [109]. The potency of autophagy cellular markers was significantly enhanced by the attachment of γ-tocotrienol with oridonin [110]. Simultaneously, this potency also improved the transformation of LC3-I to LC3-II, Atg3, Atg7, Beclin-1, Atg5-Atg12, LAMP-1, and cathepsin-D.

Geraniol: Fruits, vegetables, and cereal grains contain geraniol, a secondary metabolite, that in human PC-3 prostate cancer cells was shown to induce apoptosis and Atg cell death [111].

Seriniquinone: Seriniquinone is a medicinal compound derived from a marine bacterium of the genus Serinicoccus. It exhibits effective anti-proliferative activity against melanoma cell lines by activation of autophagocytosis [112].

Thymoquinone: Thymoquinone (TQ) is a key bioactive natural product isolated from black cumin, *Nigella sativa* L. TQ is related to increased volumes of autophagic vacuoles, as well as LC3 proteins, and enhances the combination of autophagosomes in the head and neck squamous carcinoma cells [113]. TQ effectively reduced the growth rate of GBM cells in association with the lysosomal inhibitor CQ, mostly causing apoptosis, and initiating autophagy [114]. LC3-II and p62 proteins are enhanced by the expression of TQ [114]. It was also shown that TQ prevented the growth of irinotecan-inhibited LoVo colon cancer cells by primarily activating apoptosis before autophagy [115]. Activation of the p38 and JNK MAP kinase pathways is related to TQ, which causes autophagic cell death [115]. TQ-induced caspase and autophagic cell death were also related to mitochondrial outward membrane penetrability [114].

Magnolol: Magnolol, isolated from *Magnolia officinalis,* exhibited anti-cancer and anti-tumor activity by directing autophagy and apoptosis signaling pathways [116]. However, magnolol promotes autophagy-mediated cell death in human non-small lung cancer H460 cells at high concentrations [117]. Additionally, in SGC-7901 human gastric adenocarcinoma cells, magnolol has been shown to repress PI3K/Akt pathway and encourage death by inducing autophagy [118].

Polyphenol mulberry extract: Mulberry is derived from *Morus Alba* leaf, which modulates the autophagy AMPK/PI3K/Akt pathway [119]. In Hep3B human hepatocellular carcinoma cells, mulberry extract modulates autophagic or apoptotic cell death by the activation of the p53-dependent pathway [120]. It has been found that in NB cells, *Morus alba* root extract initiates apoptosis through FOXO-Caspase-3 dependent pathway [121].

Oblongifolin C: Oblongifolin C (OC), isolated from *Garcinia yunnanensis* Hu, is a strong autophagy inhibitor [122]. Treatment with OC resulted in an augmented number of autophagosomes as well as reduced SQSTM1/p62 and degradation [123]. OC showed anticancer effectiveness by improved staining of LC3 puncta, SQSTM1, and cleaved CASP-3, along with decreased expression levels of lysosomal cathepsins in an in vivo xenograft mouse model [122].

Naphthazarin: Naphthazarin repressed the Akt/PI3K pathway by activating autophagy and apoptosis pathways and by inducing A549 lung cancer cell death [124].

Sulforaphane: Sulforaphane is derived from cruciferous vegetables, such as cauliflower, cabbage, broccoli, and hoary weed [125]. In human prostate cancer PC-3 cells, sulforaphane has been shown to prompt autophagosome formation as well as acidic vesicular organelle formation [126]. Sulforaphane augmented the protein levels of LC3B-I in addition to encouraging LC3B-II processing. Exposure to sulforaphane in tumor cells displayed LC3B-II puncta related to autophagosome formation [127]. It also disrupted BECN1/BCL-2 interaction, causing autophagy initiation through the release of BECN1. Phosphorylated AKT-Ser473 levels decreased by sulforaphane in addition to simultaneous treatment with autophagy inhibitors, 3-MA or CQ, and inhibited tumor cell growth and proliferation [128]. Sulforaphane initiation decreases oxidative stress in addition to misfolded protein accumulation [129].

Mollugin: A bioactive phytochemical, mollugin, sequestered from *Rubia cordifolia* L., showed anticancer potential against numerous cancer cells [130]. Additional studies confirmed that the mTOR and ERK pathways are involved in mollugin-prompted autophagy in addition to apoptosis [131].

Jujuboside B: Jujuboside B is a saponin found in the seeds of *Zizyphus jujubavar* Spinose. It increased autophagy and apoptosis in human HCT-116 and AGS gastric adenocarcinoma cells in vitro and successfully repressed tumor growth in a xenograft model of nude mice in vivo [132]. Furthermore, jujuboside B activated p38/JNK-mediated autophagy induction through cytoplasmic vacuole formation in addition to LC3-I/II conversion [132].

Cyclovirobuxine D: Cyclovirobuxine D (CVB-D) has been shown to activate autophagy in the human MCF-7 breast cancer cell line through the addition of autophagosomes as well as raised LC3 puncta, along with augmented conversion of LC3-I to LC3-II [133]. However, CVB-D-mediated autophagy induction and cell viability were blocked by 3-MA [133].

Polygonatum odoratum lectin: Polygonatum odoratum lectin (POL) has been shown to reveal apoptosis-inducing and anti-proliferative properties in a wide range of cancer cells [134]. In human breast cancer NSCLC A549 and MCF-7 cells, POL prompted both autophagy and apoptosis [135]. POL stimulated apoptosis by preventing the AKT/NF-κB pathway, whereas augmented autophagy through AKT-mTOR pathway suppression [135].

Resveratrol: Resveratrol, a polyphenol compound derived from berries, red grapes, and peanuts, has been shown to facilitate cell death in a diverse variety of cancer cells through autophagy, apoptosis, and necrosis [136,137]. In ovarian cancer cells, resveratrol has been found to activate autophagic cell death [138]. However, resveratrol inhibited NF-κB stimulation in association with augmented permeability of lysosomal in cervical cancer cells, demonstrating autophagic cell death [139]. Additionally, in HL-60 promyelocytic leukemia cells, it has been described that resveratrol-prompted apoptosis increased LC3-II levels in addition to dependency on mTOR pathway [140]. Meanwhile, resveratrol-mediated apoptosis was related to reduced p53 and AMPK/mTOR autophagy signaling pathways in renal carcinoma cells [141]. The antiproliferative action of resveratrol has been found in liver myofibroblasts by preventing PDGF signaling and decreasing EGF-dependent DNA synthesis [142]. Additionally, in retinal endothelial cells, resveratrol has been found to increase miR15a expression under conditions of high glucose levels and decreased insulin signaling [143].

Quercetin: Quercetin, a bioflavonoid, is generally found in fruits, beverages, and vegetables. In vitro and in vivo studies have shown that it exhibits antiproliferative activities in numerous tumors conventionally related to its antioxidant properties [144].

Rottlerin: Rottlerin, a natural product sequestered from *Mallotus philippinensis*, repressed PI3K/mTOR signaling in human pancreatic cancer stem cells, and activated autophagy-mediated apoptotic cell death [145].

Angelicin: Angelicin, a psoralen, is derived from *Angelica polymorpha* and has been shown to induce apoptosis [146,147] and autophagy [148]. Angelicin increased autophagy-related proteins Atg3, Atg7 and Atg12-5 with phosphorylation of mTOR [148,149].

Ursolic acid: A triterpenoid phytochemical, ursolic acid (UA), has been found to activate autophagy by inducing LC3 in addition to p62 protein accumulation [150]. In HCT15 cells, UA repressed growth by modulating autophagy in the JNK pathway [151]. However, UA induced cytotoxicity in addition to repressing concentration-dependent TC-1 cervical cancer cells by controlling Atg5 and LC3-II, which demonstrated its anticancer activity [150]. Furthermore, in MCF7 breast cancer cells, UA activated autophagy under ER stress [152]. In PC3 prostate cancer cells, UA-induced autophagy was facilitated by the Akt/mTOR and Beclin-1 pathways [153].

Polygonatum Cyrtonema Lectin: Polygonatum Cyrtonema Lectin (PCL) inhibits the PI3K-Akt pathway in addition to prompting autophagy and apoptosis in cancer cells [154]. Although PCL significantly prevented the growth of A375 human melanoma cells, normal melanocytes were not affected [155]. Additionally, PCL has been found to stimulate both autophagy and apoptosis in A375 cells [156].

Silibinin: Silibinin, derived from *Silybum marianum*, has been shown to induce autophagic HT1080 cell death in human fibrosarcoma through diverse mechanisms, including inhibition of MEK/ERK and PI3K/Akt pathways [157].

Plant extract induces autophagy: Several plant-derived extracts from *Dioscorea nipponica* Makino, *Melandrium firmum*, marine algae, and natural flavonoids have been found to induce apoptosis and autophagic properties [158,159,160,161]. *Saussurea lappa* ethanol extract has been investigated to activate apoptosis in NB [162] and autophagy in LNCaP prostate cancer cells [163].

Ginsenoside Rk1: Ginsenoside Rk1, an NMDA receptor inhibitor [164], exhibited antitumor and autophagy modulation activities in HepG2 cells. Rk1-prompted autophagy was recognized through LC3-I to LC3-II conversion, which incorporated lysosomes into the autolysosome process [165]. However, the autophagy inhibitor, Beclin 1 siRNA or bafilomycin A1, and Rk1 combination boosted autophagic activities in HepG2 [165].

Climacostol: In mouse B16-F10 melanoma cells, autophagosomes accumulate with dysfunctional autophagic degradation. However, climacostol provides mechanistic insights and promotes autophagosome turnover through the p53-AMPK pathway and activated p53 protein levels [166].

## 5. Role of Natural Compounds in Autophagy Modulation of Neurodegenerative Diseases

Natural plant-derived compounds are able to modulate autophagy and can be used to develop treatments for neurodegenerative diseases such as Alzheimer’s disease (AD), Parkinson’s disease (PD), Huntington’s disease (HD), Amyotrophic lateral sclerosis (ALS), Spinocerebellar ataxia (SCA), and NB [167]. Oleuropein aglycone (OLE), a natural phenol, can protect the Aβ-driven cytotoxicity of neuronal cells by stimulating autophagy found in in vitro and in vivo examinations, thereby easing Aβ toxicity clearance of AD. [168,169]. Arctigenin, derived from *Arctium lappa* (L.), inhibits Aβ production by decreasing BACE1 and boosts Aβ clearance by increasing autophagy via AKT/mTOR inhibition [170]. Another polyphenol, resveratrol, significantly represses Aβ aggregation via autophagy and shows antioxidant effects in AD [171,172,173]. Kaempferol, a flavone, increases autophagy and encourages mitochondrial damage repair, protecting neuronal cells against death via rotenone-induced toxicity in PD models [174]. *Uncaria rhynchophylla* contains alkaloids of oxindole and was found to prompt Beclin-1-dependent as well as mTOR-independent autophagy, which stimulated α-synuclein clearance in the *Drosophila* model of A53T, both wild-type and mutant [175]. Conophylline, *Tavertaemontana divaricate-*derived vinca alkaloid, has been shown to promote α-synuclein degradation via the mTOR-independent pathway in MPP^+^-mediated neurotoxin [176]. Curcumin, a polyphenol from *Curcuma longa* turmeric plant, has been shown to exhibit neuroprotective properties against PD models via mTOR-dependent autophagy and preventing oxidative stress, inflammation, and α-synuclein aggregation. [177]. Trehalose, resveratrol, and onjisaponin B improve autophagic degradation of α-synuclein when treating MPTP-induced mice, in addition to stimulating AMPK and sirtuin 1 protein [178,179]. A natural alkaloid, harmine, has been found to stimulate α-synuclein removal by PKA-mediated PD pathogenesis of UPS activation [180]. Distinct AD, PD, and HD neurodegenerative syndromes might be treated via natural compounds through modulation of autophagy [1]. *Nelumbo nucifera* contains neferine, which has been shown to protect Htt mutant proteins via autophagy triggering with the AMPK/mTOR pathway [181]. Aggregate-prone mutant Htt protein has been removed by trehalose through an mTOR-independent autophagy pathway [182]. Berberine, an isoquinoline alkaloid, has been shown to prevent mutant Htt protein accumulation and activate autophagy in mouse and cell models of HD [183]. In particular, trehalose has been found to stimulate mTOR-independent autophagy in superoxide dismutase mutant mice in the management of ALS in vivo [184]. Trehalose and its analogs, such as lactulose and melibiose, have been determined to substantially reduce the aggregation of abnormal ataxin-3 by stimulation of autophagy, which reduces free radical production [185], indicating that natural compounds might be used as a therapeutic approach to control SCA.

## 6. Effect of Natural Compounds on Solid Tumors and Lymphomas

Neuroblastoma, the most common extracranial solid malignant tumor of childhood, is the third most common form of pediatric malignancy worldwide. Didymin, a dietary flavonoid glycoside, has been found to be effective in NB treatment [186]. A *Morus alba* root extract compound, Oxy R, has been shown to accumulate ROS as well as inducing autophagic and apoptotic cell death in NB cells via the FOXO and caspase-3 induction pathways [51,121]. A psoralen compound angelicin derived from *Angelica polymorpha* increases cytotoxicity in addition to inducing apoptosis through anti-apoptotic proteins (Bcl-xL, Bcl-2, and Mcl-1) downregulation in SH-SY5Y human NB cells [146,147]. However, *Saussurea lappa* Clarke, *Dioscorea nipponica* Makino, and *Melandrium firmum* extracts have been shown to have apoptotic and anti-proliferative effects in NB cells [158,162,187]. Additionally, curcumin, propolis, quercetin, icariin, withaferin A, tetrandrine, and resveratrol have been used for the treatment of aggressive GBM [188]. In addition, a combination therapy with temozolomide and resveratrol substantially diminished the growth of GBM cells in a xenograft model [189]. Natural compounds such as PTX, wortmannin, and beta-lapachone have been found to be involved in the ARF/MDM2-MDMX/p53 signaling pathway in the human retinoblastoma (RB) Y79 cell line [190].

Lymphoma is the most common type of blood cancer, and various natural compounds have been used to treat lymphomas. Several natural compounds have shown anticancer effects on lymphomas (Table 2). *Nigella sativa* Linn. containing TQ has been shown to induce apoptosis and ROS production at 5 and 10 mM concentrations of 24 h treatment in ABC-DLBCL activated B cell lymphoma cell lines [191]. It has been stated that fuxocanthinol induces apoptosis in addition to arrest in cell cycle progression in lymphoma cells [192].

Curcumin has been reported to induce antitumor effects on CH12F3 cell lines through DNA breaks and apoptosis [193]. An extract from *Peperomia tetraphylla*, Peperobtusin A, has been found to induce S phase cell cycle arrest and p38 MAPK-dependent apoptosis in U937 cells [194]. However, induction of apoptosis, necrosis, and cell cycle arrest by psilostachyin C has been found in lymphoma BW5147 cell lines [196]. In a xenograft mouse model, 11(13)-dehydroivaxillin (DHI) has shown antitumor activity via the apoptosis pathway [195].

## 7. Therapeutic View of Autophagy in Heart/Cardiovascular Diseases

Autophagy has been shown to play a major regulatory role in cellular longevity [197] and in heart/cardiovascular diseases towards maintaining homeostasis by conserving cardiac function and structure [198] and determining a potential connection between ventricular fibrillation and autophagy [199]. There is evidence suggesting that mitochondrial autophagy downregulation plays a vital role in mitochondrial dysfunction, as well as heart failure, while re-establishment of mitochondrial autophagy mitigates this heart dysfunction [200]. It has been revealed that cardiac myocytes are reduced through autophagy inhibitor, 3-MA, with AMP-activated protein kinase (AMPK) activation via inactivation of mTOR under glucose deprivation [201]. In addition, the autophagy-related gene, Beclin-1, protected the heart in LPS-induced sepsis in a mouse model and targeted autophagy induction, and therefore has an important role with therapeutic potential for treating cardiovascular diseases [202]. Under high glucose conditions, it has been found that heme-oxygenase-1 (HO-1) overexpression via human HO-1 recombinant plasmid inhibited dysfunction of cardiac arrest, improving autophagy levels. Numerous hemin concentrations induce HO-1, which affects endothelial cell mitochondrial dysfunction. Likewise, exposure to hemin further prompts mitophagy, although it is not adequate to inhibit cell death [203]. It has been found that regulating Sirt3 concentration in myocytes via proper autophagy expression in diverse stages of myocardial ischemia-reperfusion might effectively decrease the morbidity of patients with myocardial infarction in the future [204]. Metformin has been used to treat cardiotoxicity induced by doxorubicin via the autophagy pathway [205]. In contrast, a non-selective β-adrenergic agonist, isoproterenol, has been investigated for the treatment of autophagy as well as cell death pathway induced cardiac injury such as necrosis or apoptosis induced cardiac injury [206]. Currently, it is accepted that pharmacological approaches targeting autophagy and apoptosis are useful in the treatment of coronary heart disease, therefore providing new favorable therapeutic interventions in the future [207]. Hence, consideration of emerging molecular mechanisms towards the understanding of autophagy has recently gained importance for regulating cellular systems that can be used to prevent and treat cardiovascular disease [208].

## 8. Perspectives of Naturally Occurring Autophagy Modulators in Cancer Therapy

Despite anti-cancer drug development, natural compounds contribute to cancer and cancer stem cell [209] suppressive action without any severe complications, and thus it is an emerging topic with respect to autophagy. Natural compounds have also appeared as unique therapeutic agents for cancer and drug repositioning via their impact on autophagy [210]. It is remarkable that the study of cell fate by natural stimulators should be the focal point, although technological dissimilarities in autophagy detection may affect the outcome. In addition, in vivo examination of natural stimulators of autophagy in cancer therapy or prevention will be an emerging field of research in the future. Moreover, studies on the effects of natural product-derived molecules on autophagy should be highlighted at the translational level.

## 9. Conclusions and Future Directions

Cancer cells can convert and reuse necessary amino acids by imposing autophagic signaling pathways, thereby ensuring their consistent growth and longevity. Accordingly, autophagy serves a dual role in cancer—augmentation or hindrance—and can either boost or down-regulate cancer cell proliferation. As a result, autophagy moderators can play the role of a promising novel therapeutic approach towards cancer inhibition by helping overcome resistance against chemotherapy and radiotherapy. Potential targets of autophagic progress and autophagosome formation in the initial phase and lysosomal deterioration in the late phase have been considered. However, mTOR and AMPK are the leading regulatory molecules of autophagy in upstream signaling pathways. These are also known targets for natural compound derived modulators of autophagy. Therefore, using plant-derived and semisynthetic components to target the regulation of the entire pathway can provide an opportunity to design a powerful remedy for cancer patients.

## Figures and Tables

**Figure 1 biomedicines-08-00517-f001:**
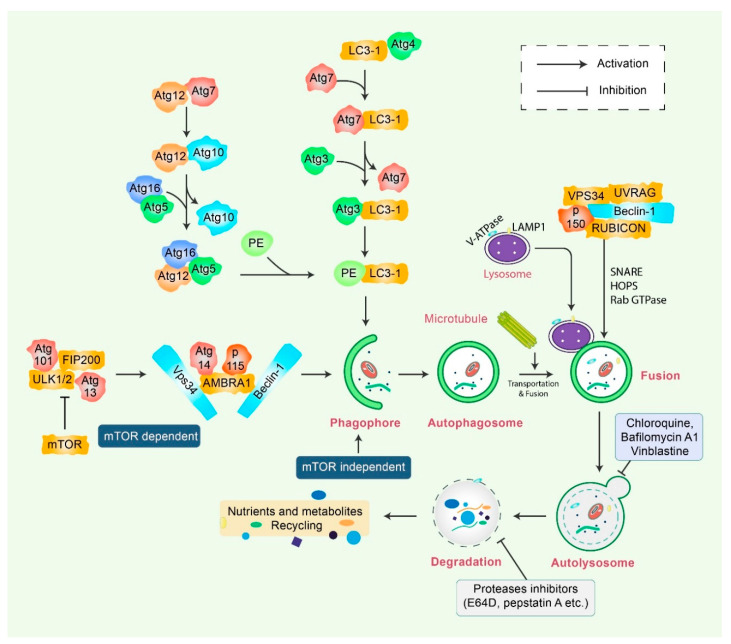
Molecular mechanism of autophagy signaling. Autophagy is generally initiated by a deficiency of growth factors or nutrients that trigger AMPK or mTOR inhibition. This stimulates FIP200 and Atg13 associated ULK complex. After that phosphorylation of Beclin-1, it leads to the activation of VPS34, which further initiates the formation of phagophore. Conjugation of Atg5-Atg12 encompasses Atg7 as well as Atg10 to form an Atg12-Atg5-Atg16 complex that also stimulates phagophores formation. Atg5, in addition to Atg12, makes an Atg16 complex, which acts as an E3-function concerning LC3-PE assembly (LC3-II). This complex likewise initiates the formation of a phagophore. Specifically, LC3-II is an autophagy marker that is ultimately interrupted via autolysosomes. Maturation of autophagosome leads to fusion with lysosomes in association with several lysosomal proteins, finally leading to degradation of cargo in addition to recycling of metabolites and nutrients.

**Figure 2 biomedicines-08-00517-f002:**
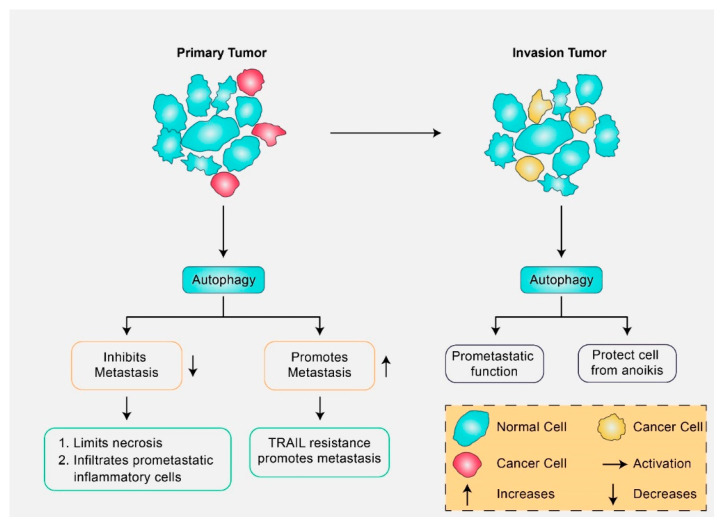
In cancer cells, autophagy exhibits a dual role in metastasis and invasion. Depending on the stage and type of tumor cell, autophagy can prevent or stimulate cancer cell maintenance. In the early stages of autophagy in primary tumors, metastasis has been suppressed via anti-metastatic elements in addition to stimulation of tumor conservation along with progression by inducing cellular resistance against TRAIL, a tumor necrosis factor-related apoptosis-inducing ligand, to induce apoptosis. Later, in the course of invasion, tumor cells spread and enter lymphatic or blood vessels. Here, autophagy shows a pro-metastatic function through cell protection from anoikis triggered via extracellular matrix detachment.

**Figure 3 biomedicines-08-00517-f003:**
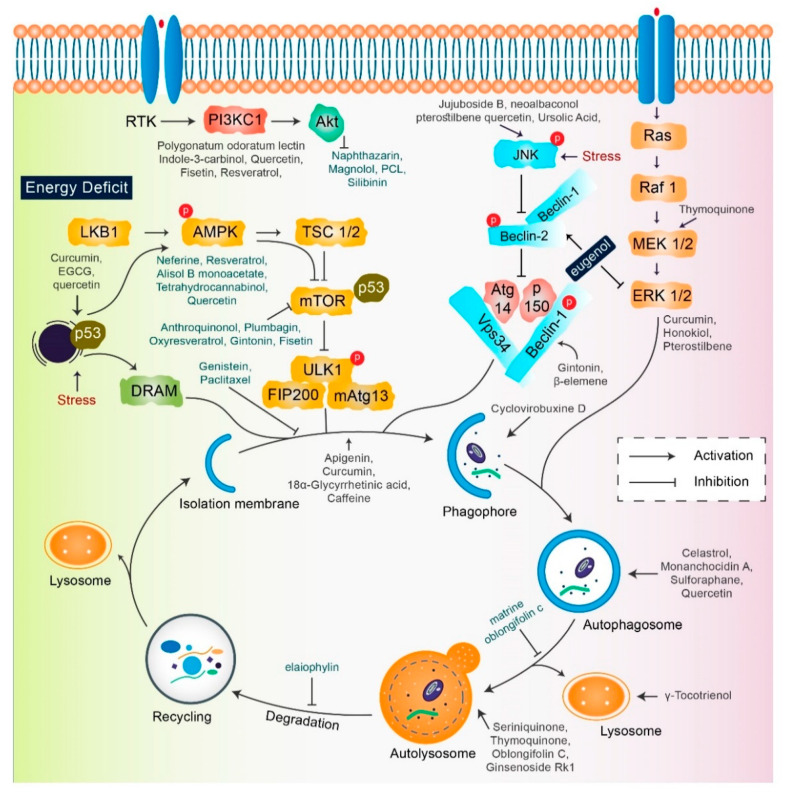
Key autophagy pathway and its regulators (natural compounds) for treating diverse cancers. This figure is modified from Wang et al. [64]. Natural compounds activate autophagy by several autophagic signaling targets and show a close relationship with cancer development and control. Multiple signaling pathways are activated or inhibited by natural compounds to modulate cancer cells.

**Figure 4 biomedicines-08-00517-f004:**
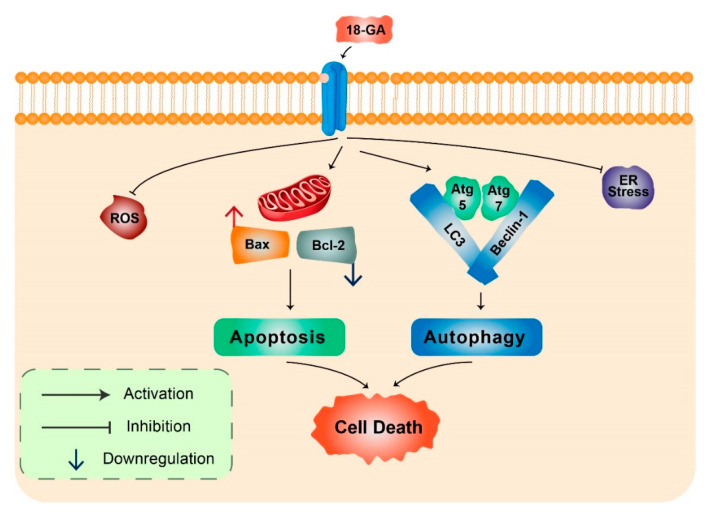
18α-Glycyrrhetinic acid-mediated cell death in human neuroblastoma cells. 18-GA induces mitochondria-mediated apoptosis to alter mitochondrial membrane potential in a significant change in with Bax/Bcl-2 ratio. Autophagy-related proteins Atg5/7, Beclin-1, LC3, and p62 were activated during 18-GA-mediated autophagic cell death. Beclin-1 contributes to autophagy in addition to apoptosis induction in 18-GA-mediated cell death. Additionally, ROS and ER stress showed no changes during 18-GA treatment in SH-SY5Y and B103.

**Figure 5 biomedicines-08-00517-f005:**
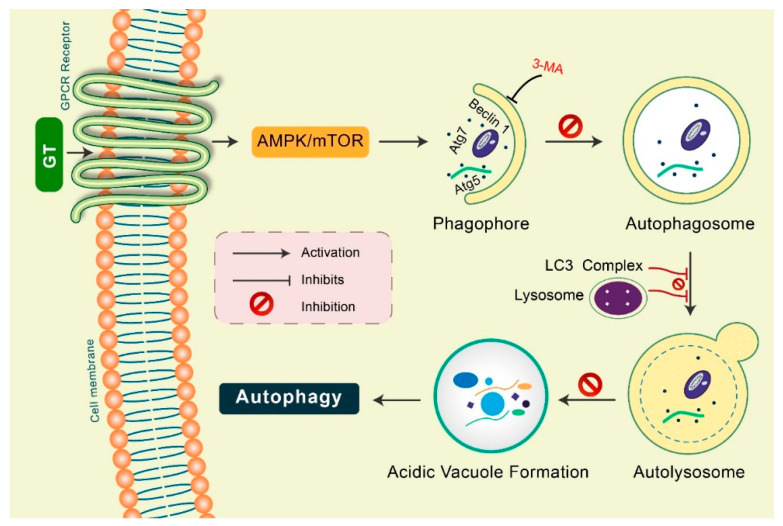
Gintonin (GT) activates autophagy in mouse cortical astrocytes. GT-mediated autophagy was dependent on GPCR-LPA receptors. Pretreatment with an LPA receptor antagonist, Ki16425, considerably decreased GT-mediated autophagy, while LPA agonist pretreatment, LPA 18:1, augmented autophagy in astrocytes. Autophagy initiation proteins Atg5, Atg7, and Beclin-1 were found to be activated in GT-mediated autophagy in astrocytes. Inhibition of autophagy by 3-MA, GT-mediated autophagy was prevented. Pretreated with E-64d/pepstatin A and bafilomycin A1, GT additionally improved LC3 puncta formation indicating enhanced autophagic flux in cortical astrocytes. GT treatment increased acidic vacuole formation along with p62 protein accumulation during this autophagy process.

**Figure 6 biomedicines-08-00517-f006:**
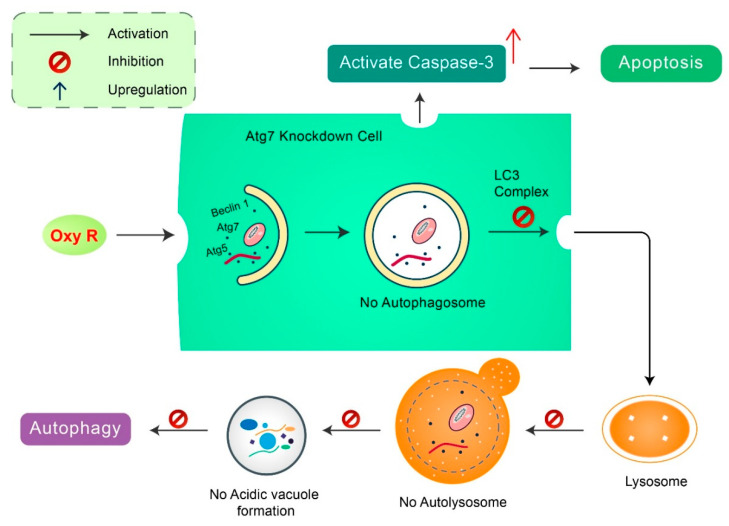
Natural compound oxyresveratrol (Oxy R) stimulates apoptosis and autophagy-dependent cell death. Oxy R activates autophagy through autophagy initiation Atg7 and Beclin 1 dependent pathways. Atg7 knockdown prevents autophagy, but activates apoptotic cell death in SH-SY5Y cells. Blockage of autophagy via 3-MA, Oxy R inhibits autophagic cell death. When apoptosis signaling is inhibited by Z-DEVD-FMK caspase-3 inhibitor, Oxy R exhibits autophagic cell death.

**Table 1 biomedicines-08-00517-t001:** Mechanism of action of combination of natural compounds and autophagy blockers, 3-methyladenine (3-MA), chloroquine (CQ), and bafilomycin A1 (Baf A1) in modulating autophagy and apoptosis in various cancer cell models.

Natural Compound/Chemical	Cell Model	Molecular Mechanisms	Combination with Autophagy Inhibitors	References
18α-Glycyrrhetinic acid	Neuroblastoma	Autophagy and apoptosis induction	CQ combination induces cell death	[42]
Resveratrol	Glioblastomas	Induction of apoptosis and autophagy via reduction of ROS/MAPK pathway	Baf A1 combination augments apoptotic cell death	[43]
Curcumin	Glioblastomas	Induction of autophagy and apoptosis	CQ combination rises apoptotic cell death	[44]
Gintonin	Cortical astrocytes and glioblastoma cells	Induction of autophagy	CQ and Baf A1 combination stimulates autophagy	[45]
Quercetin	U373MG cells	PI3K/Akt inhibition and causes apoptosis and autophagy	CQ combination increases apoptotic cell death	[46]
Paclitaxel	Human lung carcinoma A549 cells	Induction of autophagy and apoptosis	3-MA combination enhances apoptotic cell death	[47]
Genistein	MIA PaCa-2 human pancreatic cancer cell	Autophagy and apoptosis induction	CQ combination stimulates apoptotic cell death	[48]
Honokiol	Non-small cell lung cancer A549 and H460 cells	p62 and LC3 induces autophagy induction	Combination of CQ triggered apoptosis	[49]
Ginsenoside Compound K	Neuroblastoma SK-N-BE(2) and SH-SY5Y cells	ROS-mediated autophagy inhibition and apoptosis induction	Combination of CQ inhibit autophagy and induces apoptosis	[50]
Oxyresveratrol	Neuroblastoma SK-N-BE(2) and SH-SY5Y cells	PI3K/AKT/mTOR pathway independent autophagy and apoptosis	3-MA combination enhances apoptotic cell death	[51]

**Table 2 biomedicines-08-00517-t002:** Anticancer effects of natural cancers in the management of lymphomas.

Natural Compounds	Cellular/Animal Models	Concentrations/Doses	Effects and Mechanisms	References
Thymoquinone from *Nigella sativa*	Activated B cell lymphoma cell lines	5–10 mM	ROS production and apoptosis	[191]
Resveratrol	Neuroblastoma	30 µM	Apoptosis induction and antiproliferation	[137]
Fuxocanthinol	Primary effusion lymphomas BCBL-1 and TY-1	1.3–5 µM	Apoptosis induction and cell cycle arrest	[192]
Curcumin	CH12F3 lymphoma cells	5 µM	Caspase-3 dependent apoptosis and DNA damage	[193]
Peperobtusin A	Lymphoma U937 cells	25, 50, 100 μM	Caspase-3, 8, 9 dependent apoptosis and p38 MAPK activation	[194]
11(13)-dehydroivaxillin (DHI)	Lymphoid malignancies of NHL cells xenografts	5, 7, 10 μM	Induction of NF-κB and apoptosis	[195]
Psilostachyin C	Murine lymphoma cell line BW5147	0.01–50 μg/mL	Induction of apoptosis, necrosis, and ROS generation	[196]

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
