# Peer review of "Molecular Insights into the Multifunctional Role of Natural Compounds: Autophagy Modulation and Cancer Prevention"

_biomedicines, 2020, doi:10.3390/biomedicines8110517_

Round 1

Reviewer 1 Report

  1. On page 1, line 6, it would be better if the word “and” could be replaced by a superscript comma.
  2. On page 1, line 36, "it has" could be replaced by "they have".
  3. On page 2, line 46, what does stresses viz. mean?
  4. On page 5, line 152, what does 78 Atg7 mean?
  5. On page 7, figure 3, please make sure compound names are all separated by comma, especially the ones with AMPK-modulating activities.
  6. On page 10, figure 5, what is the molecular target of gintonin (GT)? What is the relationship between LPA receptors and autophagy, which is not illustrated in the figure.
  7. On page 11, figure 6, since the authors tried to mention Atg KO cell, how come Atg7 still exists in the figure?

Author Response

First of all, we would like to express our sincere gratitude for the time and effort the reviewer had put into reviewing our manuscript.

  1. On page 1, line 6, it would be better if the word “and” could be replaced by a superscript comma.

>>Response: We checked and correct it (page 1, line 5).

  1. On page 1, line 36, "it has" could be replaced by "they have".

>>Response: We replaced  it accordingly (page 2, line 40).

  1. On page 2, line 46, what does stresses viz. mean?

>>Response: We modified the sentence (page 3, line 49).

  1. On page 5, line 152, what does 78 Atg7 mean?

>>Response: This is a typographical mistake. We checked from the reference and corrected the sentence (page 8, line 176-179).

  1. On page 7, figure 3, please make sure compound names are all separated by comma, especially the ones with AMPK-modulating activities.

>>Response: We modified the figure accordingly and compound names are separated by comma (page 11, figure 3).

  1. On page 10, figure 5, what is the molecular target of gintonin (GT)? What is the relationship between LPA receptors and autophagy, which is not illustrated in the figure.

>>Response: The main molecular target of GT was induced autophagy via LPA GPCR receptor mediated signaling. We modified the figure 5. Our recently published paper described that GT-mediated autophagy is primarily depend on GPCR-LPA receptors (Ref: 87; Rahman et al. 2020). (page 15, line 306-308).

  1. On page 11, figure 6, since the authors tried to mention Atg KO cell, how come Atg7 still exists in the figure?

>>Response: This is my previously published paper (Ref: 51; Rahman et al. 2017). We want to mention here Atg7 knock down cell inhibit LC3 complex formation and activates apoptosis signaling in SH-SY5Y neuroblastoma cells. We corrected the figure by writing Atg7 knock down cell. (page 17, figure 6).

Reviewer 2 Report

This review manuscript is a very cursory summary of the effects induced by natural compounds in terms of autophagy modulation and cancer prevention in different cancer models. A major crucial issue is about English language and style, as too many inconsistencies were found and manuscript is quite unreadable in the present form. Moreover, the current description lacks many details and in some cases the information provided both in the main text and in figures is incorrect.

In addition, as a narrative review it lacks details about the paper's methodology, including how was the literature reviewed and which criteria were chosen for article inclusion, or exclusion.

Author Response

This review manuscript is a very cursory summary of the effects induced by natural compounds in terms of autophagy modulation and cancer prevention in different cancer models. A major crucial issue is about English language and style, as too many inconsistencies were found and manuscript is quite unreadable in the present form. Moreover, the current description lacks many details and in some cases the information provided both in the main text and in figures is incorrect.

In addition, as a narrative review it lacks details about the paper's methodology, including how was the literature reviewed and which criteria were chosen for article inclusion, or exclusion.

>>Response: First of all, we would like to express our sincere gratitude for the time and effort the reviewer had put into reviewing our manuscript. We reorganized and rewritten our manuscript and changed portion are marked by blue color. We massively revised and improved the quality of our manuscript by the professional English language editors (Company name: Editage, Ref#: MDATA_3).

Reviewer 3 Report

Comments to the authors (Ataur Rahman et al.,):

Molecular insights into multifunctional role of natural compounds: autophagy modulation and cancer prevention

The topic of the manuscript is especially highly valued since it is focused on the discussion of autophagy modulation and cancer prevention. Dysregulation of autophagy may lead to improvement of various cellular diseases. including cancer. Chemotherapy is able to modulate autophagy, however the side effects of recently used chemical drugs are severe. Therefore, developing new modulators of autophagy are of great interest. Different phytochemical of natural compounds and its derivatives have fascinated gain-of-attention to use favorable autophagy modulators in cancer treatment with minimal side effects. In the current review, the authors discuss the promising role of natural compounds to modulate autophagy pathway to control cancer treatment and prevention.

General comments:

The topic of the review discusses  timely and up -to-date research novelties. I highly recommend the publication of this review paper since it includes useful information in the field.

The submitted review is clearly written, with a high level of English language. The manuscript is well-organized into six chapters. However, the chapters uneven in proportion. It would be advisable to write the Introduction part in more detail and briefly summarize the essence of the research results on autophagy achieved so far.

In general, some parts of the review described in very detail, however some chapters missing important points. For example, it is very important fact that autophagy has been described as a dual function in promotion as well as inhibition of metastasis. Although this point is mentioned in the 3. chapter (Molecular mechanism of autophagy signaling in cancer pathogenesis), but it would be useful to discuss more precisely the details. It would be also interesting to write more about the effect of natural agents on tumor microenvirovment. Since microRNAs are among important regulatory molecules in cancer signalling, it would be advisible to write a little on this topic as well.

This reviewer would advise to the authors, please devote a chapter to the role of natural compounds in autophagy modulation of other diseases beyond cancer, e.g., cardiovascular or neurodegenerative disease. Also, the authors put a short summary of the effect of these compounds on solid tumors and lymphomas in a separate table.

Please include additional references, such as the following publications:

Sabari Saha et al.: Autophagy in health and disease: A comprenahsive review. Biomedicine & Pharmacotherapy. Volume 104, August 2018, Pages 485-495

Deng S, Shanmugam MK, Kumar AP, Yap CT, Sethi G, Bishayee A.Cancer. Targeting autophagy using natural compounds for cancer prevention and therapy. 2019 Apr 15;125(8):1228-1246. doi: 10.1002/cncr.31978. Epub 2019 Feb 12.PMID: 30748003 Free article. Review.

Zecchini S, Proietti Serafini F, Catalani E, Giovarelli M, Coazzoli M, Di Renzo I, De Palma C, Perrotta C, Clementi E, Buonanno F, Ortenzi C, Marcantoni E, Taddei AR, Picchietti S, Fausto AM, Cervia D. Dysfunctional autophagy induced by the pro-apoptotic natural compound climacostol in tumour cells Cell Death Dis. 2018 Dec 19;10(1):10. doi: 10.1038/s41419-018-1254-x.

Russo M, Russo GL.  Autophagy inducers in cancer. Biochem Pharmacol. 2018 Jul;153:51-61. doi: 10.1016/j.bcp.2018.02.007. Epub 2018 Feb 10.PMID: 29438677 Review.

Sohn EJ, Park HT.Cancer Cell Int. Natural agents mediated autophagic signal networks in cancer.2017 Nov 28;17:110. doi: 10.1186/s12935-017-0486-7. eCollection 2017.PMID: 29209152 Free PMC article. Review.

Juhasz B, Varga B, Gesztelyi R, Kemeny-Beke A, Zsuga J, Tosaki A. Resveratrol: a multifunctional cytoprotective molecule. Curr Pharm Biotechnol. 2010 Dec;11(8):810-8. doi: 10.2174/138920110793262079.

Additionally:

Autophagic processes may be different in prolipherated tissues (e.g., cancer cells) in comparison with nonprolipherated tissues, such as the myocardium, although the signaling processes may be very similar. Namely, the authophagic processes (autophagic cell death) may be useful at a certain degree in the myocardium, if the injured cells are able to remove themselves the damaged cell organelles. Thus, the sudden cardiac death caused by ventricular fibrillation cardiac death may be prevented. Therefore, the following articles in the view of autophagy/heart diseases are suggested to be discussed generally in the revised version of this manuscript (Shirakabe A, et al., Circulation. 2016 Mar 29;133(13):1249-63. doi: 10.1161/CIRCULATIONAHA.115.020502; Leidal AM, et al., Nat Cell Biol. 2018 Dec;20(12):1338-1348. doi: 10.1038/s41556-018-0235-8. Epub 2018 Nov 26; Sun Y, et al., Circulation. 2018 Nov 13;138(20):2247-2262. doi: 10.1161/CIRCULATIONAHA.117.032821; Meyer G, et al., Curr Pharm Des. 2013;19(39):6912-8. doi: 10.2174/138161281939131127122510; Lekli I, et al., J Cell Mol Med. 2017 Jun;21(6):1058-1072. doi: 10.1111/jcmm.13053. Zilinyi R, et al., Molecules. 2018 May 15;23(5):1184. doi: 10.3390/molecules23051184. Gyongyosi A, et al., Curr Pharm Des. 2019;25(19):2192-2198. doi: 10.2174/1381612825666190619145025. Sciarretta S, et bal., Annu Rev Physiol. 2018 Feb 10;80:1-26. doi: 10.1146/annurev-physiol-021317-121427. Epub 2017 Oct 25. Matsui Y, et al., Circ Res. 2007 Mar 30;100(6):914-22. doi: 10.1161/01.RES.0000261924.76669.36. Dong Y, et al., J Mol Cell Cardiol. 2019 Nov;136:27-41. doi: 10.1016/j.yjmcc.2019.09.001. Zheng Y, et al., J Cell Physiol. 2019 May;234(5):5488-5495. doi: 10.1002/jcp.27329. Gyongyosi A, et al., Biochem Biophys Res Commun. 2019 Apr 16;511(4):732-738. doi: 10.1016/j.bbrc.2019.02.140).

Figures:

All Figures are high quality and logically illustrate the processes of the autophagy.

Author Response

Molecular insights into multifunctional role of natural compounds: autophagy modulation and cancer prevention

The topic of the manuscript is especially highly valued since it is focused on the discussion of autophagy modulation and cancer prevention. Dysregulation of autophagy may lead to improvement of various cellular diseases. including cancer. Chemotherapy is able to modulate autophagy, however the side effects of recently used chemical drugs are severe. Therefore, developing new modulators of autophagy are of great interest. Different phytochemical of natural compounds and its derivatives have fascinated gain-of-attention to use favorable autophagy modulators in cancer treatment with minimal side effects. In the current review, the authors discuss the promising role of natural compounds to modulate autophagy pathway to control cancer treatment and prevention.

General comments:

The topic of the review discusses  timely and up -to-date research novelties. I highly recommend the publication of this review paper since it includes useful information in the field.

>>Response: First of all, we would like to express our sincere gratitude for the time and effort the reviewer had put into reviewing our manuscript.

The submitted review is clearly written, with a high level of English language. The manuscript is well-organized into six chapters. However, the chapters uneven in proportion. It would be advisable to write the Introduction part in more detail and briefly summarize the essence of the research results on autophagy achieved so far.

>>Response: We added more information in introduction part (page 3, line 66-70; page 4, line 78-80)

In general, some parts of the review described in very detail, however some chapters missing important points. For example, it is very important fact that autophagy has been described as a dual function in promotion as well as inhibition of metastasis. Although this point is mentioned in the 3. chapter (Molecular mechanism of autophagy signaling in cancer pathogenesis), but it would be useful to discuss more precisely the details. It would be also interesting to write more about the effect of natural agents on tumor microenvirovment. Since microRNAs are among important regulatory molecules in cancer signaling, it would be advisible to write a little on this topic as well.

>>Response: We added more information in effect of natural agents on tumor microenvirovment (Page 6, line 132-136; page 16, line 334-337; page 18, line 364-367; page 21, line 456-458) and role of microRNAs in cancer signaling(Page 6, line 137-145; page 12, line 233-234; page 16, 337-339; page 18, line 368-371; page 21, line 458-460).

This reviewer would advise to the authors, please devote a chapter to the role of natural compounds in autophagy modulation of other diseases beyond cancer, e.g., cardiovascular or neurodegenerative disease. Also, the authors put a short summary of the effect of these compounds on solid tumors and lymphomas in a separate table.

>>Response: We added two more chapter and one table. Here, we added cardiovascular  and neurodegenerative disease:

chapter 5. Role of natural compounds in autophagy modulation of neurodegenerative disease (page 23, line 497-530).

Chapter 6. Effect of natural compounds on solid tumors and lymphomas (page 24, line 531-552; table 2, page 25).

Chapter 7. Therapeutic view of autophagy in heart/cardiovascular diseases (page 22, line 561-587).

Please include additional references, such as the following publications:

Sabari Saha et al.: Autophagy in health and disease: A comprenahsive review. Biomedicine & Pharmacotherapy. Volume 104, August 2018, Pages 485-495

>>Response: Added in page 23, reference 170.

Deng S, Shanmugam MK, Kumar AP, Yap CT, Sethi G, Bishayee A. Cancer. Targeting autophagy using natural compounds for cancer prevention and therapy. 2019 Apr 15;125(8):1228-1246. doi: 10.1002/cncr.31978. Epub 2019 Feb 12.PMID: 30748003 Free article. Review.

>>Response: Added in page 18, reference 111.

Zecchini S, Proietti Serafini F, Catalani E, Giovarelli M, Coazzoli M, Di Renzo I, De Palma C, Perrotta C, Clementi E, Buonanno F, Ortenzi C, Marcantoni E, Taddei AR, Picchietti S, Fausto AM, Cervia D. Dysfunctional autophagy induced by the pro-apoptotic natural compound climacostol in tumour cells Cell Death Dis. 2018 Dec 19;10(1):10. doi: 10.1038/s41419-018-1254-x.

>>Response: Added in page 23, line 494-496, reference 169.

Russo M, Russo GL.  Autophagy inducers in cancer. Biochem Pharmacol. 2018 Jul;153:51-61. doi: 10.1016/j.bcp.2018.02.007. Epub 2018 Feb 10.PMID: 29438677 Review.

>>Response: Added in page 9, line 188-189, reference 52.

Sohn EJ, Park HT.Cancer Cell Int. Natural agents mediated autophagic signal networks in cancer.2017 Nov 28;17:110. doi: 10.1186/s12935-017-0486-7. eCollection 2017.PMID: 29209152 Free PMC article. Review.

>>Response: Added in page 26, reference 213.

Juhasz B, Varga B, Gesztelyi R, Kemeny-Beke A, Zsuga J, Tosaki A. Resveratrol: a multifunctional cytoprotective molecule. Curr Pharm Biotechnol. 2010 Dec;11(8):810-8. doi: 10.2174/138920110793262079.

>>Response: Added in page 23, page 505-507, reference 176.

Additionally:

Autophagic processes may be different in prolipherated tissues (e.g., cancer cells) in comparison with nonprolipherated tissues, such as the myocardium, although the signaling processes may be very similar. Namely, the authophagic processes (autophagic cell death) may be useful at a certain degree in the myocardium, if the injured cells are able to remove themselves the damaged cell organelles. Thus, the sudden cardiac death caused by ventricular fibrillation cardiac death may be prevented. Therefore, the following articles in the view of autophagy/heart diseases are suggested to be discussed generally in the revised version of this manuscript (Shirakabe A, et al., Circulation. 2016 Mar 29;133(13):1249-63. doi: 10.1161/CIRCULATIONAHA.115.020502; Leidal AM, et al., Nat Cell Biol. 2018 Dec;20(12):1338-1348. doi: 10.1038/s41556-018-0235-8. Epub 2018 Nov 26; Sun Y, et al., Circulation. 2018 Nov 13;138(20):2247-2262. doi: 10.1161/CIRCULATIONAHA.117.032821; Meyer G, et al., Curr Pharm Des. 2013;19(39):6912-8. doi: 10.2174/138161281939131127122510; Lekli I, et al., J Cell Mol Med. 2017 Jun;21(6):1058-1072. doi: 10.1111/jcmm.13053. Zilinyi R, et al., Molecules. 2018 May 15;23(5):1184. doi: 10.3390/molecules23051184. Gyongyosi A, et al., Curr Pharm Des. 2019;25(19):2192-2198. doi: 10.2174/1381612825666190619145025. Sciarretta S, et bal., Annu Rev Physiol. 2018 Feb 10;80:1-26. doi: 10.1146/annurev-physiol-021317-121427. Epub 2017 Oct 25. Matsui Y, et al., Circ Res. 2007 Mar 30;100(6):914-22. doi: 10.1161/01.RES.0000261924.76669.36. Dong Y, et al., J Mol Cell Cardiol. 2019 Nov;136:27-41. doi: 10.1016/j.yjmcc.2019.09.001. Zheng Y, et al., J Cell Physiol. 2019 May;234(5):5488-5495. doi: 10.1002/jcp.27329. Gyongyosi A, et al., Biochem Biophys Res Commun. 2019 Apr 16;511(4):732-738. doi: 10.1016/j.bbrc.2019.02.140).

>>Response: We added new chapter ‘7. Therapeutic view of autophagy in heart/cardiovascular diseases’ using the above mentioned articles. (page 25, line 561-587, references: 200-211).

Figures:

All Figures are high quality and logically illustrate the processes of the autophagy.

>>Response: We are grateful to reviewer wonderful comments regarding figures.

Round 2

Reviewer 2 Report

The review manuscript has been formally reviewed and in the present form is clear e nice. Moreover, additional data from literature have been provided, thus improving value and soundness of the paper.  However, a few minor points to be addressed still remain in order to render the paper suitable for publication:

  1. As a narrative review it still lacks details about the paper's methodology, including how was the literature reviewed and which criteria were chosen for article inclusion, or exclusion;
  2. Please, carefully check your English also in Figures and Tables, because some inconsistencies and typos still remain (e.g.: in Figure 2, change "TRIAL" and correct "limit necrosis";  in Table 1, correct "Induction of apoptosis and autophagy via reduces ROS/MAPK pathway", and so on)

Author Response

The review manuscript has been formally reviewed and in the present form is cleare nice. Moreover, additional data from literature have been provided, thus improving value and soundness of the paper.  However, a few minor points to be addressed still remain in order to render the paper suitable for publication:

>>Response: First of all, we would like to express our sincere gratitude for the time and effort the reviewer had put into reviewing our manuscript.

  1. As a narrative review it still lacks details about the paper's methodology, including how was the literature reviewed and which criteria were chosen for article inclusion, or exclusion;

>>Response: A literature search as well as theoretical and methodological contributions of the molecular mechanism of natural compounds in autophagy modulation and cancer prevention were conducted using PubMed, Scopus, Google Scholar, and Google that includes all original research articles written in English on multifunctional role of natural compounds. Searching was conducted from January 2020 using various keywords including autophagy, natural compounds, cancer, phytochemical. neurodegenerative diseases, solid tumors and lymphomas, heart/cardiovascular diseases, perspectives role autophagy in cancer therapy and so on. All figures were generated using Adobe Illustrator software.

Additionally, we chosen this topic because our first author Dr. Md. Ataur Rahman, last 5 years, working on natural compounds, autophagy, cancer, and neurodegenerative diseases. Dr. Rahman has several research articles in the selected topic ‘natural compounds: autophagy modulation and cancer prevention’ which we already cited in this manuscript and are listed below:

[1] M.A. Rahman, H. Rhim, Therapeutic implication of autophagy in neurodegenerative diseases, BMB Rep 50 (2017) 345-354.

[15] M.A. Rahman, M.R. Rahman, T. Zaman, M.S. Uddin, R. Islam, M.M. Abdel-Daim, H. Rhim, Emerging Potential of Naturally Occurring Autophagy Modulators Against Neurodegeneration, Curr Pharm Design 26 (2020) 772-779.

[42] M.A. Rahman, K. Bishayee, K. Habib, A. Sadra, S.O. Huh, 18alpha-Glycyrrhetinic acid lethality for neuroblastoma cells via de-regulating the Beclin-1/Bcl-2 complex and inducing apoptosis, Biochem Pharmacol 117 (2016) 97-112.

[45] M.A. Rahman, H. Hwang, S.Y. Nah, H. Rhim, Gintonin stimulates autophagic flux in primary cortical astrocytes, Journal of Ginseng Research 44 (2020) 67-78.

[51] M.A. Rahman, K. Bishayee, A. Sadra, S.O. Huh, Oxyresveratrol activates parallel apoptotic and autophagic cell death pathways in neuroblastoma cells, Biochim Biophys Acta Gen Subj 1861 (2017) 23-36.

[62] M.A. Rahman, M.S. Rahman, M.J. Uddin, A.N.M. Mamum-Or-Rashid, M.G. Pang, H. Rhim, Emerging risk of environmental factors: insight mechanisms of Alzheimer's diseases, Environ Sci Pollut R (2020).

[140] M.A. Rahman, N.H. Kim, S.H. Kim, S.M. Oh, S.O. Huh, Antiproliferative and Cytotoxic Effects of Resveratrol in Mitochondria-Mediated Apoptosis in Rat B103 Neuroblastoma Cells, Korean J Physiol Pha 16 (2012) 321-326.

[149] M.A. Rahman, N.H. Kim, H. Yang, S.O. Huh, Angelicin induces apoptosis through intrinsic caspase-dependent pathway in human SH-SY5Y neuroblastoma cells, Mol. Cell. Biochem. 369 (2012) 95-104.

[150] M.A. Rahman, K. Bishayee, S.O. Huh, Angelica polymorpha Maxim Induces Apoptosis of Human SH-SY5Y Neuroblastoma Cells by Regulating an Intrinsic Caspase Pathway, Mol Cells 39 (2016) 119-128.

[161] M.A. Rahman, H. Yang, N.H. Kim, S.O. Huh, Induction of apoptosis by Dioscorea nipponica Makino extracts in human SH-SY5Y neuroblastoma cells via mitochondria-mediated pathway, Anim Cells Syst 18 (2014) 41-51.

[162] M.A. Rahman, H. Yang, S.S. Lim, S.O. Huh, Apoptotic Effects of Melandryum firmum Root Extracts in Human SH-SY5Y Neuroblastoma Cells, Exp Neurobiol 22 (2013) 208-213.

[212] M.A. Rahman, S.K. Saha, M.S. Rahman, M.J. Uddin, M.S. Uddin, M.G. Pang, H. Rhim, S.G. Cho, Molecular Insights Into Therapeutic Potential of Autophagy Modulation by Natural Products for Cancer Stem Cells, Front Cell Dev Biol 8 (2020). ARTN 283.

  1. Please, carefully check your English also in Figures and Tables, because some inconsistencies and typos still remain (e.g.: in Figure 2, change "TRIAL" and correct "limit necrosis"; in Table 1, correct "Induction of apoptosis and autophagy via reduces ROS/MAPK pathway", and so on)

>>Response: We massively checked the English. This typographical mistake correct accordingly (page 5, figure 2).

We also corrected ‘Induction of apoptosis and autophagy via reduction of ROS/MAPK pathway’ (page 6, table 1). Additionally, we corrected some words in table 2 (page 16, table 2).